# INDUCTIVE COLLABORATIVE FILTERING VIA RELATION GRAPH LEARNING

## ABSTRACT

Collaborative filtering has shown great power in predicting potential user-item ratings by factorizing an observed user-item rating matrix into products of two sets of latent factors. However, the user-specific latent factors can only be learned in transductive setting and a model trained on existing users cannot adapt to new users without retraining the model. In this paper, we propose an inductive collaborative filtering framework that learns a hidden relational graph among users from the rating matrix. We first consider a base matrix factorization model trained on one group of users' ratings and devise a relation inference model that estimates their underlying relations (as dense weighted graphs) to other users with respect to historical rating patterns. The relational graphs enable attentive message passing from users to users in the latent space and are updated in end-to-end manner. The key advantage of our model is the capability for inductively computing user-specific representations using no feature, with good scalability and superior expressiveness compared to other feature-driven inductive models. Extensive experiments demonstrate that our model achieves state-of-the-art performance for inductive learning on several matrix completion benchmarks, provides very close performance to transductive models when given many training ratings and exceeds them significantly on cold-start users.

## 1 INTRODUCTION

As information explosion has become one major factor affecting human life in the decade, recommender systems, which can filter useful information and contents of user's potential interests, play an increasingly indispensable part in day-to-day activities. Recommendation problems can be generally formalized as matrix completion (MC) where one has a user-item rating matrix whose entries, which stand for interactions of users with items (ratings or click behaviors), are partially observed. The goal of MC is to predict missing entries (unobserved or future potential interactions) in the matrix based on the observed ones.

Modern recommender systems need to meet two important requirements in order for desirable effectiveness and practical utility. First of all, *recommendation models should have enough expressiveness to capture diverse user interests and preferences so that the systems can accomplish personalized recommendation*. Existing methods based on collaborative filtering (CF) or, interchangeably, matrix factorization (MF) have shown great power in this problem by factorizing the rating matrix into two classes of latent factors (i.e., embeddings) for users and items respectively, and further leverage dot-product of two factors to predict potential ratings (Koren et al., 2009; Rendle et al., 2009; Srebro et al., 2004; Zheng et al., 2016b). Equivalently, for each user, the methods consider a one-hot user index as input, assume a user-specific embedding function (which maps a user index to a latent factor), and use the learnable latent factor to represent user's preferences in a low-dimensional space. One can select proper dimension size to control balance between capacity and generalization. Recent works extend MF with complex architectures, like multi-layer perceptrons (Dziugaite & Roy, 2015), recurrent units (Monti et al., 2017), autoregressive models (Zheng et al., 2016a), graph neural networks (van den Berg et al., 2017), etc., and achieve state-of-the-art results on most benchmarks.

The second requirement stems from a key observation from real-world scenarios: *recommender systems often interact with a dynamic open world where new users, who are not exposed to models during training, may appear in test stage*. This requires that models trained on one group of users

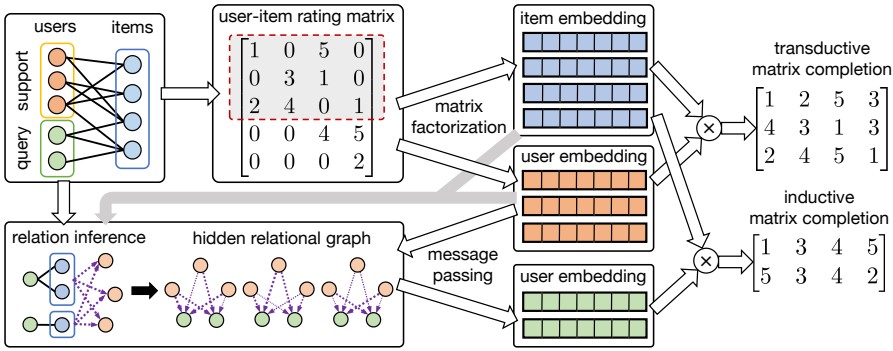

Figure 1: Model framework of inductive relational matrix factorization.

manage to adapt to unseen users. However, the above-mentioned CF models would fail in this situation since the user-specific embeddings are parametrized for specific users and need to be learned collaboratively with all other users in transductive setting. One brute-force way is to retrain the whole model with an augmented rating matrix, but extra time cost would be unacceptable for online systems. There are quite a few studies that propose inductive matrix completion models using user features (Jain & Dhillon, 2013; Xu et al., 2013; Cheng et al., 2016; Ying et al., 2018; Zhong et al., 2018). Their different thinking paradigm is to target a user-sharing mapping from user features to user representations, instead of from one-hot user indices used by CF models. Since the feature space is shared among users, such methods are able to adapt a model trained on existing users to unseen users. Nevertheless, feature-driven models often suffer from limited expressiveness with low-quality features that have weak correlation with target labels. For example, users with the same age and occupation (commonly used features) may have distinct ratings on movies and music. Unfortunately, high-quality features that can unveil user interests for personalized recommendation are often hard to collect due to increasingly concerned privacy issues.

A following question arises: *Can we have a recommendation model that guarantees enough expressiveness for personalized recommendation and enables inductive learning?* In fact, to simultaneously meet the two requirements is a non-trivial challenge when high-quality user features are unavailable. First, to achieve either of them, one needs to compromise on the other. In fact, the one-hot user indices (together with learnable user-specific embeddings) give a maximized capacity for learning distinct user preferences from historical rating patterns. To make inductive learning possible, one needs to construct a shared input feature space among users out of the rating matrix, as an alternative to one-hot user indices. However, the new constructed features have relatively insufficient expressive power. Second, the computation based on new feature space often bring extra costs for time and space, which limits model's scalability to large-scale datasets.

In this paper, we propose an inductive collaborative filtering model (IRCF) [1] as a general CF framework that achieves inductive learning for matrix completion and meanwhile guarantees enough expressiveness and scalability. As shown in Fig. 1, we consider a base transductive matrix factorization model trained on one group of users (called support users) and a relation inference model that aims to estimate their relations to another group of users (called query users) w.r.t. historical rating patterns. The (multiple) estimated relational graphs enable attentively message passing from users to users in the latent space and compute user-specific representations in an inductive way. The output user representations can be used to compute product with item representations to predict ratings in the matrix, as is done by CF models. Compared with other methods, one key advantage of IRCF is the capability for inductively computing user-specific representations without using features.

Besides, our method possesses the following merits. 1) **Expressiveness:** A general version of our model can minimize reconstruction loss to the same level as matrix factorization under a mild condition. Also, we qualitatively show its superior expressiveness than feature-driven and local-graph-based inductive models that may fail in some typical cases. Empirically, IRCF provides very close performance to transductive CF models when given sufficient training ratings. 2) **Generalization:** IRCF manages to achieve state-of-the-art results on new (unseen) users compared with inductive models. Also, IRCF gives much better accuracy than transductive models when training data becomes sparse and outperforms other competitors in extreme cold-start recommendation. 3) **Scal-**

---

[1]The codes will be released.

**ability:** Our model enables mini-batch training and efficient inference. In experiments, IRCF is averagely ten times faster than local-graph-based inductive model. 4) **Flexibility:** As a general CF framework, IRCF is flexible to incorporate with various architectures (e.g. MLP-based, GNN-based, autoregressive, etc.) as a base transductive model as well as deal with implicit user feedbacks.

## 2 BACKGROUND AND RELATED WORKS

In this section, we present some background of the problem and discuss relationships to related works. We consider a general matrix completion (MC) problem which deals with a user-item rating matrix $R = \{r_{ui}\}_{M \times N}$ where $M$ and $N$ are the numbers of users and items, respectively. For implicit feedback, $r_{ui}$ is a binary entry which denotes whether user $u$ rated (or clicked on, reviewed, liked, purchased, etc.) item $i$ or not. For explicit feedback, $r_{ui}$ records rating value of user $u$ on item $i$. The entries of $R$ are partially observed and the goal is to estimate the missing values in the matrix[2]. In the following, we introduce related works and highlight their differences to our paper. In Fig. 5, we provide an illustration for comparison with these methods.

**General Collaborative Filtering.** Existing methods for MC are generally based on collaborative filtering (CF) or, interchangeably, matrix factorization (MF) where user $u$ (resp. item $i$) corresponds to a $d$-dimensional latent factor (i.e., one-hot embedding) $\mathbf{p}_u$ (resp. $\mathbf{q}_i$).Then one has a prediction model $\hat{r}_{ui} = f(\mathbf{p}_u, \mathbf{q}_i)$ where $f$ can be basically specified as simple dot product or some complex architectures, like neural networks, graph neural networks, etc. One advantage of CF models is that the user-specific embedding $\mathbf{p}_u$ (as learnable parameters) can provide enough expressive power for learning diverse personal preferences from user historical behaviors and decent generalization ability through collaborative learning with all the users and items. Furthermore, the user embeddings possess rich profile information as representation of user preferences and can benefit various downstream tasks, like target advertisement, user-controllable recommendation (Ma et al., 2019; Cen et al., 2020), influence maximization (Khalil et al., 2017; Manchanda et al., 2019), friend recommendation, etc. However, such user-specific embedding limits the model in transductive learning and when it comes to new users during test, one has to retrain the model (often associated with embeddings for both new and existing users as well as items) with new augmented rating matrix. Admittedly, one can consider local updates for embeddings of new users with fixed item embeddings learned from existing users. Nevertheless, such operation makes model learning for users independent from each other, which is prone for over-fitting compared with collaborative learning in CF models. Besides, it requires 'incremental' learning for each new user, while IRCF can deliver on-the-fly inference in online systems.

**Feature-driven Recommendation.** The CF models do not require any side information other than the rating matrix, but cannot be trained inductively due to the learnable user-specific embedding $\mathbf{p}_u$. To address the issue, one can leverage side information such as attribute features to achieve inductive learning. Define user features (like age, occupation, etc.) as $\mathbf{a}_u$ and item features (like movie genre, director, etc.) as $\mathbf{b}_i$. The feature-driven model targets a prediction model $\hat{r}_{ui} = g(\mathbf{a}_u, \mathbf{b}_i)$. Since the space of $\mathbf{a}_u$ is shared among users, a model trained on one group of users can adapt to other users without retraining. However, feature-driven models often provide limited performance since the shared feature space is not expressive enough compared to one-hot embedding space. Another issue is that high-quality features are hard to collect in practice. We note that our model does not require user features for inductive collaborative filtering.

**Inductive Matrix Completion.** There are a few existing works that attempt to handle inductive matrix completion using only user-item rating matrix. (Hartford et al., 2018) (F-EAE) puts forward an exchangeable matrix layer that takes a whole rating matrix as input and inductively outputs prediction for missing ratings. However, the scalability of F-EAE is limited since it requires the whole rating matrix as input for training and inference for users, while IRCF enables mini-batch training and efficient inference. Besides, (Zhang & Chen, 2020) (IGMC) proposes to use local subgraphs of user-item pairs in a bipartite graph of rating information as input features and further adopt graph neural networks as representation tool to encode subgraph structures for rating prediction. The model achieves inductive learning via replacing users' one-hot index embeddings by shared input features (i.e., index-free local subgraph structures). Differently, IRCF maintains the ability to give

---

[2]Also, in some situations, one targets a ranking list of items for each user as top-N recommendation. In this paper, we focus on predicting missing values in the matrix and leave top-N recommendation for future works.

user-specific embeddings, which represent users' preferences and interests and can be used for downstream tasks (like target advertisement, user-controllable recommendation, influence maximization, etc.) while F-EAE and IGMC can merely output a prediction score without such user representation. Moreover, the expressiveness of IGMC is limited since the local subgraph structures can be indistinguishable for users with distinct behaviors (see Section 3.2 for more discussions), while IRCF has equivalent expressiveness as original CF models.

**Item-based CF Models.** Some previous works use item embeddings as representation for users. (Cremonesi et al., 2010; Kabbur et al., 2013) proposes to use a combination of items rated by users to compute user embeddings and frees the model from learning parametric one-hot embeddings for users. Furthermore, there are quite a few auto-encoder architectures for recommendation problem, leveraging user's rating vector (ratings on all the items) as observed input, estimate user embedding (as latent variables) based on that, and output prediction for missing values in the rating vector (Sedhain et al., 2015; Liang et al., 2018). With item embeddings and user's rating history, these methods achieve inductive learning for users and can adapt to new users during test. On methodological level, IRCF has the following differences: 1) IRCF learns to use weighted combination of users' embeddings to compute embeddings for new users, and such combination weights possess interpretability for underlying social influence or user proximity; 2) IRCF considers both users' and items' one-hot embeddings in general CF models that maintain better capacity than item-based CF models that only considers learnable parameters in item embedding space.

## 3 METHODOLOGY

We propose a new inductive collaborative filtering model without using features. We borrow the general idea from graph neural networks (GNNs) which define a computational graph over nodes and edges and further harness message passing through edges to obtain new node representations that aggregate neighbored information. Our high-level methodology stems from a key observation: there exist a (or multiple) latent relational graph among users that represents preference proximity and behavioral interactions[3]. Based on the relational graphs, we can leverage message passing, propagating learned embeddings from one group of users to others, especially, in an inductive way.

We formulate our model through two sets of users: *support users* (denoted by $\mathcal{U}_1$), for which we learn their embeddings in transductive setting, and *query users* (denoted by $\mathcal{U}_2$), for which we consider message passing to inductively compute their embeddings. Assume $|\mathcal{U}_1| = M_1$ and $|\mathcal{U}_2| = M_2$. Correspondingly, we have two rating matrices: $R_1 = \{r_{ui}\}_{M_1 \times N}$ (given by $\mathcal{U}_1$) and $R_2 = \{r_{u'i}\}_{M_2 \times N}$ (given by $\mathcal{U}_2$). Note that there is no strict requirement for $\mathcal{U}_1$ and $\mathcal{U}_2$ in our formulation. Two common settings are $\mathcal{U}_1 \cap \mathcal{U}_2 = \emptyset$, where query users are distinct from support users, and $\mathcal{U}_1 = \mathcal{U}_2$, where support users equal to query users in training. We consider both cases in Section 4.

We train a (transductive) matrix factorization model for $\mathcal{U}_1$ using $R_1$, denoted as $\hat{r}_{ui} = f_\theta(\mathbf{p}_u, \mathbf{q}_i)$, where $\mathbf{p}_u \in \mathbb{R}^d$ denotes user-specific embedding for user $u$ in $\mathcal{U}_1$, $\mathbf{q}_i \in \mathbb{R}^d$ denotes item-specific embedding for item $i$ and $f_\theta$ can be simple dot-product or a neural network with parameter $\theta$. In Appendix C, we present details for two specifications for $f_\theta$ using neural network and graph convolution network, which are used in our implementation. Denote $\mathbf{P}_1 = \{\mathbf{p}_u\}_{M_1 \times d}$, $\mathbf{Q} = \{\mathbf{q}_i\}_{N \times d}$ and the objective function becomes

$$\min_{\mathbf{P}_1, \mathbf{Q}, \theta} \mathcal{D}_{\mathcal{S}_1}(\hat{R}_1, R_1), \tag{1}$$

where $\hat{R}_1 = \{\hat{r}_{ui}\}_{M_1 \times N}$, $\mathcal{D}_{\mathcal{S}_1}(\hat{R}_1, R_1) = \frac{1}{T_1} \sum_{(u,i) \in \mathcal{S}_1} l(r_{ui}, \hat{r}_{ui})$ and $\mathcal{S}_1 \in ([M_1] \times [N])^{T_1}$ is a set with size $T_1$ containing indices of observed entries in $R_1$. Here one can use cross-entropy or square loss for $l(r_{ui}, \hat{r}_{ui})$.

The key question is how to learn the relational graph among users and consider message passing for inductive representation learning. We next propose our inductive relation inference model.

---

[3]Social networks and following networks in social media can be seen as realizations of such relational graphs, but in most cases, the graphs are unobserved and implicitly affect user's decisions and behaviors.

### 3.1 INDUCTIVE RELATION INFERENCE MODEL

Assume $\mathbf{C} = \{c_{uu'}\}_{M_1 \times M_2}$, where $c_{uu'} \in \mathbb{R}$ denotes weighted edge from user $u \in \mathcal{U}_1$ to user $u' \in \mathcal{U}_2$, and define $\mathbf{c}_{u'} = [c_{1u'}, c_{2u'}, \cdots c_{M_1 u'}]^\top$ the $u'$-th column of $\mathbf{C}$. Then we express embedding of user $u'$ as $\mathbf{p}_{u'} = \mathbf{c}_{u'}^\top \mathbf{P}_1$, the weighted sum of embeddings of support users. The rating can be predicted by $\hat{r}_{u'i} = f_\theta(\mathbf{p}_{u'}, \mathbf{q}_i)$ and the problem can be formulated as

$$\min_{\mathbf{C}, \mathbf{Q}} \mathcal{D}_{\mathcal{S}_2}(\hat{R}_2, R_2), \tag{2}$$

where $\hat{R}_2 = \{\hat{r}_{u'i}\}_{M_2 \times N}$, $\mathcal{D}_{\mathcal{S}_2}(\hat{R}_2, R_2) = \frac{1}{T_2} \sum_{(u',i) \in \mathcal{S}_2} l(r_{u'i}, \hat{r}_{u'i})$ and $\mathcal{S}_2 \in ([M_2] \times [N])^{T_2}$ is a set with size $T_2$ containing indices of observed entries in $R_2$. The essence of above method is taking attentive pooling as message passing from support to query users. We first justify this idea by analyzing its capacity and then propose a parametrized model that enables it for inductive learning.

**Theoretical Justification** If we use dot-product for $f_\theta$ in the MF model, then we have $\hat{r}_{u'i} = \mathbf{p}_{u'}^\top \mathbf{q}_i$. We compare (2) with using matrix factorization over $R_2$:

$$\min_{\mathbf{P}_2, \mathbf{Q}} \mathcal{D}_{\mathcal{S}_2}(\hat{R}_2, R_2), \tag{3}$$

where $\mathbf{P}_2 = \{\mathbf{p}_{u'}\}_{M_2 \times d}$ and have the following theorem.

**Theorem 1.** *Assume (3) can achieve $\mathcal{D}_{\mathcal{S}_2}(\hat{R}_2, R_2) < \epsilon$ and the optimal $\mathbf{P}_1$ given by (1) satisfies column-full-rank, then there exists at least one solution for $\mathbf{C}$ in (2) such that $\mathcal{D}_{\mathcal{S}_2}(\hat{R}_2, R_2) < \epsilon$.*

The only condition that $\mathbf{P}_1$ is column-full-rank can be trivially guaranteed since $d \ll N$. The theorem shows that the proposed model can minimize the reconstruction loss of MC to at least the same level as matrix factorization which gives maximized capacity for learning personalized user preferences from historical rating patterns.

**Parametrization** We showed that using attentive pooling does not sacrifice model capacity than CF models under a mild condition. However, directly optimizing over $\mathbf{C}$ is intractable due to its $O(M_1 M_2)$ parameter space and $\mathbf{c}_{u'}$ is user-specific which disallows inductive learning. Hence, we parametrize $\mathbf{C}$ with an attention network, significantly reducing parameters and enabling it for inductive learning. Concretely, we estimate the adjacency score between user $u'$ and user $u$ as

$$c_{u'u} = \frac{\mathbf{e}^\top [\mathbf{W}_q \mathbf{d}_{u'} || \mathbf{W}_k \mathbf{p}_u]}{\sum_{u_0 \in \mathcal{U}_1} \mathbf{e}^\top [\mathbf{W}_q \mathbf{d}_{u'} || \mathbf{W}_k \mathbf{p}_{u_0}]}, \tag{4}$$

where $\mathbf{e} \in \mathbb{R}^{2d \times 1}$, $\mathbf{W}_q \in \mathbb{R}^{d \times d}$, $\mathbf{W}_k \in \mathbb{R}^{d \times d}$ are trainable parameters, $||$ denotes concatenation and $\mathbf{d}_{u'} = \sum_{i \in \mathcal{I}_{u'}} \mathbf{q}_i$. Here $\mathcal{I}_{u'} = \{i | r_{u'i} > 0\}$ includes the historically rated items of user $u'$. The attention network captures first-order user proximity on behavioral level and also maintains second-order proximity that users with similar historical ratings on items would have similar relations to other users. Besides, if $\mathcal{I}_{u'}$ is empty (for extreme cold-start recommendation), we can randomly select a group of items or use (the embedding of) user's attribute features as $\mathbf{d}_{u'}$ if features are available. We provide details in Appendix D.

The normalization in (4) requires computation for all the support users, which limits scalability to large dataset. Therefore, we use sampling strategy to control the size of support users in relation graph for each query user and further consider multi-head attentions that independently sample different subsets of support users. The attention score given by the $l$-th head is

$$c_{u'u}^{(l)} = \frac{(\mathbf{e}^{(l)})^\top [\mathbf{W}_q^{(l)} \mathbf{d}_{u'} || \mathbf{W}_k^{(l)} \mathbf{p}_u]}{\sum_{u_0 \in \mathcal{U}_1^{(l)}} (\mathbf{e}^{(l)})^\top [\mathbf{W}_q^{(l)} \mathbf{d}_{u'} || \mathbf{W}_k^{(l)} \mathbf{p}_{u_0}]}, \tag{5}$$

where $\mathcal{U}_1^{(l)}$ denotes a subset of support users sampled from $\mathcal{U}_1$. Each attention head independently aggregates embeddings of different subsets of support users and the final inductive representation for user $u'$ can be given as

$$\mathbf{p}_{u'} = \mathbf{W}_o \left[ \sum_{u \in \mathcal{U}_1} c_{u'u}^{(1)} \mathbf{W}_v^{(1)} \mathbf{p}_u || \sum_{u \in \mathcal{U}_1} c_{u'u}^{(2)} \mathbf{W}_v^{(2)} \mathbf{p}_u || \cdots || \sum_{u \in \mathcal{U}_1} c_{u'u}^{(L)} \mathbf{W}_v^{(L)} \mathbf{p}_u \right], \tag{6}$$

where $\mathbf{W}_o \in \mathbb{R}^{d \times Ld}$ and $\mathbf{W}_v^{(l)} \in \mathbb{R}^{d \times d}$. To keep the notation clean, we denote $\mathbf{p}_{u'} = h_w(\mathbf{d}_{u'})$ and $w = \cup_{l=1}^{L} \{\mathbf{e}^{(l)}, \mathbf{W}_q^{(l)}, \mathbf{W}_k^{(l)}, \mathbf{W}_v^{(l)}\} \cup \{\mathbf{W}_o\}$.

There exist some common thinkings as for the rationals of our model and inductive graph representation learning (Hamilton et al., 2017; Ying et al., 2018; Zhang et al., 2018), which considers message passing from existing nodes to new nodes over graphs. Differently, IRCF jointly estimate neighbored nodes for a target node (given by attention scores) and learn node representations based on that, while the latter often assumes a given observed graph and directly learn node representations. Furthermore, IRCF can deal with nodes with new users with no historical edge, while the latter would fail for new nodes with no observed edge (if without node attribute features).

**Optimization** The training process consists of two contiguous stages. First, we pretrain a MF model in transductive setting via (1) and obtain embeddings $\mathbf{P}_1$, $\mathbf{Q}$ and network $f_\theta$. Second, we train our relation model $h_w$ with fixed $\mathbf{P}_1$, $\mathbf{Q}$ via

$$\min_{w,\theta} \mathcal{D}_{\mathcal{S}_2}(\hat{R}_2, R_2). \tag{7}$$

We found that using fixed $\mathbf{Q}$ in the second stage contributes to much better performance on test ratings of query users than optimizing over it.

We further analyze the generalization ability of inductive model on query users. Also, consider $f_\theta$ as dot-product operation and we assume $c_{u'u} \in \mathbb{R}^+$ to simplify the analysis. In the next theorem, we show that the generalization error $\mathcal{D}(\hat{R}_2, R_2) = \mathbb{E}_{(u',i)}[l(r_{u'i}, \hat{r}_{u'i})]$ on query users would be bounded by the numbers of support users and observed ratings of query users.

**Theorem 2.** *Assume that 1) $\mathcal{D}$ is L-Lipschitz, 2) for $\forall \hat{r}_{u'i} \in \hat{R}_2$ we have $|\hat{r}_{u'i}| \leq B$, and 3) the L1-norm of $\mathbf{c}_{u'}$ is bounded by $H$. Then with probability at least $1 - \delta$ over the random choice of $\mathcal{S}_2 \in ([M_2] \times [N])^{T_2}$, it holds that for any $\hat{R}_2$,*

$$\mathcal{D}(\hat{R}_2, R_2) \leq \mathcal{D}_{\mathcal{S}_2}(\hat{R}_2, R_2) + O\left( 2LHB\sqrt{\frac{2M_2 \ln M_1}{T_2}} + \sqrt{\frac{ln(1/\delta)}{T_2}} \right). \tag{8}$$

The theorem shows that a smaller size of $\mathcal{U}_1$ would make the generalization error bound tighter. Looking at both Theorem 1 and 2, we will find that the configuration of $\mathcal{U}_1$ has an important effect on model capacity and generalization ability. On one hand, we need to make support users in $\mathcal{U}_1$ 'representative' of diverse user behavior patterns on item consumption in order to guarantee enough model capacity. Also, we need to control the size of $\mathcal{U}_1$ in order to maintain generalization ability. Based on these insights, how to properly select support users can be an interesting direction for future investigation. Our experiments in Section 4 provide more interesting discussions on this point.

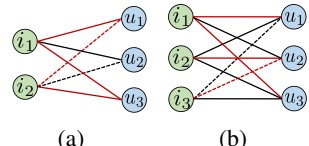

(a)  (b)

Figure 2: Feature-driven and local-graph-based models fail in (a) and (b), respectively. IRCF works in both cases with superior expressiveness.

### 3.2 DISCUSSIONS

To shed more lights on model expressiveness, we provide a comparison with feature-driven and local-graph-based inductive models through cases in Fig. 2. Here we assume ratings are within $\{-1, 1\}$ (positive, denoted by red line, and negative, denoted by black line). The solid lines are observed ratings for training and dash lines are test ratings. In Fig. 2 (a), local-graph-based models can give right prediction relying on different 1-hop local subgraphs of $(u_1, i_2)$ and $(u_2, i_2)$, while feature-driven models will fail once $u_1$ and $u_2$ have the same features. In Fig. 2 (b), local-graph-based models will fail since $(u_1, i_3)$ and $(u_2, i_3)$ possess the same 1-hop local subgraphs though with distinct ground-truth ratings on $i_3$. By contrast, CF models and IRCF can recognize that $u_3$ has similar rating patterns with $u_1$ and different from $u_2$, thus pushing the embedding of $u_1$ (resp. $u_2$) close to (resp. distant from) $u_3$, which guides the model to right prediction. Note that the first case becomes a common issue when the feature space is small while the second case becomes general when the rating patterns of users are not distinct enough throughout a dataset, which induces similar local subgraph structures. In short, IRCF achieves inductive learning and maintains as good expressiveness as transductive models.

Table 1: RMSEs for all the users (All), query users (Query) and new users (New) in different datasets. We highlight the best scores among all the (resp. inductive) models with bold (resp. underline).

| Method | Inductive | Feature | Douban | | | ML-100K | | | ML-1M | | |
|---|---|---|---|---|---|---|---|---|---|---|---|
| | | | All | Query | New | All | Query | New | All | Query | New |
| PMF | No | No | 0.737 | 0.718 | - | 0.932 | 1.003 | - | 0.851 | 0.946 | - |
| NNMF | No | No | 0.729 | 0.712 | - | 0.925 | 0.987 | - | 0.848 | **0.940** | - |
| GCMC | No | No | 0.731 | **0.710** | - | 0.911 | 0.989 | - | **0.838** | 0.941 | - |
| NIMC | Yes | Yes | 0.772 | 0.745 | 0.766 | 1.015 | 1.065 | 1.089 | 0.873 | 0.995 | 1.059 |
| PinSAGE | Yes | Yes | 0.769 | 0.743 | 0.763 | 1.008 | 1.055 | 1.083 | 0.859 | 0.961 | 1.055 |
| BOMIC | Yes | Yes | 0.735 | 0.713 | 0.764 | 0.931 | 1.001 | 1.088 | 0.847 | 0.953 | 1.057 |
| F-EAE | Yes | No | 0.738 | - | - | 0.920 | - | - | 0.860 | - | - |
| IGMC | Yes | No | **0.721** | 0.718 | 0.743 | **0.905** | 0.997 | 1.031 | 0.857 | 0.956 | 0.997 |
| **IRCF-NN** | Yes | No | 0.731 | 0.712 | 0.749 | 0.931 | 0.999 | 1.057 | 0.844 | 0.952 | 0.991 |
| **IRCF-GC** | Yes | No | 0.732 | 0.712 | **0.719** | **0.905** | **0.981** | **0.999** | 0.839 | 0.944 | **0.956** |

Table 2: RMSEs (resp. AUCs) on query users in ML-10M (resp. Amazon-Books).

| Method | ML-10M ($\downarrow$ better) | Amazon ($\uparrow$ better) |
|---|---|---|
| PMF | 0.928 | 0.917 |
| NNMF | 0.922 | 0.921 |
| GCMC | **0.919** | 0.922 |
| IRCF-NN | 0.924 | 0.942 |
| IRCF-GC | **0.919** | **0.948** |

Table 3: RMSEs on cold-start users in ML-1M using features.

| Method | ML-1M ($\downarrow$ better) |
|---|---|
| Wide&Deep | 1.0260 |
| GCMC | 0.9688 |
| AGNN | 1.0087 |
| MeLU | 0.9625 |
| IRCF-HY | **0.9367** |

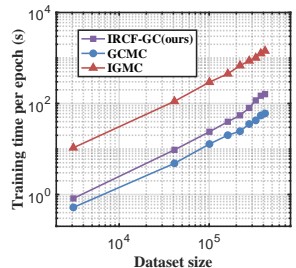

Figure 3: Scalability test in ML-1M.

## 4 EXPERIMENTS

In this section, we conduct experiments to verify the proposed model. We consider five benchmark datasets *Douban*, *Movielens-100K (ML-100K)*, *Movielens-1M (ML-1M)*, *Movielens-10M (ML-10M)* and *Amazon-Books*. The statistics of datasets and more details are in Appendix E.1. For Douban and ML-100K, we use the training/test split provided by (Monti et al., 2017), which is used by (van den Berg et al., 2017; Hartford et al., 2018; Zhang & Chen, 2020). For ML-1M and ML-10M, we also follow previous works and use 9:1 training/test spliting. For Amazon-Books with implicit user feedbacks, we use the last ten ratings of each user for test and the remaining for training.

We consider two specifications for $f_\theta$ in our model: IRCF-NN, which adopts multi-layer perceptron for $f$ (following the design of NNMF (Dziugaite & Roy, 2015)), and IRCF-GC, which uses graph convolution network for $f$ (following the design of GCMC (van den Berg et al., 2017)). The details are in Appendix C. We also extend IRCF with user attribute features as IRCF-HY, a hybrid model that takes both user one-hot indices and features as input (see details in Appendix D.1), for cold-start recommendation. IRCF-HY considers one-hot/multi-hot embeddings for various features and further a neural network for prediction, following the design of Wide&Deep network (Cheng et al., 2016). In Appendix E.2, we present detailed information for hyper-parameter settings. We leave out 5% training data as validation set for parameter tuning and use early stoping strategy in training. For each dataset we run experiments with five different random seeds and report average results (the improvements in the paper are all significant so we omit confidence intervals for results).

### 4.1 COMPARATIVE EXPERIMENTS

There can be different configurations for support users and query users. To simulate real-world scenarios where new users are often with fewer historical ratings, we divide users into two sets: users with more than $\delta$ training ratings, denoted as $\overline{\mathcal{U}}_1$, and users with less than $\delta$ training ratings, denoted as $\overline{\mathcal{U}}_2$. We basically set $\delta = 30$ for Douban and three Movielens datasets, and $\delta = 20$ for Amazon-Books. Then we consider two situations: 1) we set $\mathcal{U}_1 = \overline{\mathcal{U}}_1$ and $\mathcal{U}_2 = \overline{\mathcal{U}}_2$, i.e., we treat $\overline{\mathcal{U}}_1$ as support users and $\overline{\mathcal{U}}_2$ as query users; 2) we set $\mathcal{U}_1 = \mathcal{U}_2 = \overline{\mathcal{U}}_1$, i.e., we treat $\overline{\mathcal{U}}_1$ as both

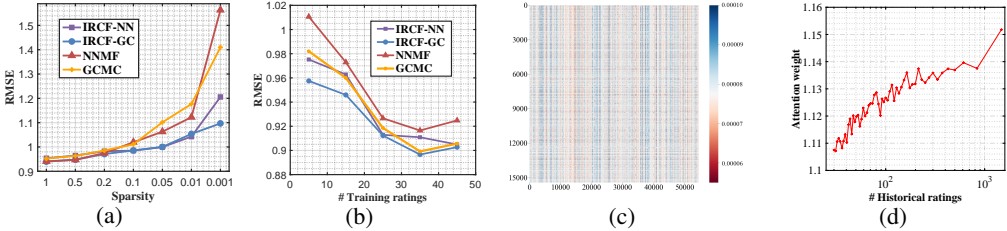

Figure 4: (a) Overall RMSE w.r.t # sparsity ratio. (b) User specific RMSE w.r.t # user's training ratings.(c) Attention weights of query users (y-axis) on support users (x-axis). (d) Support uses' accumulated attention weights w.r.t. # historical ratings.

support and query users[4]. In the first case, we can use test performance on $\overline{\mathcal{U}}_2$ to investigate model's expressiveness in inductive learning; in the second case, we can use test performance on $\overline{\mathcal{U}}_2$ to study model's generalization on new unseen users during test. Other split ways like random split or using different $\delta$'s and their impact on model performance are thoroughly discussed in Appendix F.

**Comparison with transductive & inductive models** We compare with three state-of-the-art transductive methods PMF (Salakhutdinov & Mnih, 2007), NNMF (Dziugaite & Roy, 2015), GCMC (van den Berg et al., 2017), three inductive feature-driven methods NIMC (Zhong et al., 2018), PinSAGE (Ying et al., 2018), BOMIC (Ledent et al., 2020), and two recently proposed inductive methods without using features F-EAE (Hartford et al., 2018) and IGMC (Zhang & Chen, 2020). For these competitors, we use all the training data for training. In Table 1, we report test RMSEs for all the users, inductive learning on query users and generalization on new users in Douban, ML-100k and ML-1M. As we can see, IRCF-NN gives very close RMSEs on all the users and query users to NNMF in three datasets while IRCF-GC performs nearly as well as GCMC. The empirical results prove that our inductive model possesses the same expressive power as corresponding transductive model. Compared with inductive methods, IRCF-GC achieves the best RMSEs for query users in three datasets. The results demonstrate the superior expressiveness of proposed inductive model comapred to other feature-driven and local-graph-based inductive models. Besides, for new unseen users, our model IRCF-GC gives state-of-the-art performance, achieving significant improvement over the best competitor, quantitatively 3.2% on Douban, 3.1% on ML-100K and 4.1% on ML-1M.

In Table 2, we compare IRCF-NN and IRCF-GC with PMF, NNMF and GCMC in ML-10M and Amazon-Books. Since the two datasets have no feature, we cannot train feature-driven methods. Also, F-EAE and IGMC are both hard to scale to such large datasets. In Amazon-Books, we replace square loss in our objective as cross-entropy loss and harness AUC as metric to align with implicit feedbacks. We can see that our inductive models achieve very similar RMSEs in ML-10M as transductive models, and even significantly outperform transductive models in Amazon-Books. In fact, Amazon dataset is a very sparse one with rating density 0.0012. One implication here is that our inductive model can provide better performance than transductive models for users with few ratings.

**Comparison with cold-start recommendation models** We also wonder if our inductive model can handle extreme cold-start users who have no historical rating. Note that cold-start users are different and more challenging compared to new (unseen) users. For new users, the model can still use observed historical ratings as input during test, though it cannot be trained on these ratings. In order to enable cold-start recommendation, we leverage attribute features in Movielens-1M. We use the dataset provided by (Lee et al., 2019), which contain attribute features and split warm-start and cold-start users, and follow its evaluation protocol: using the *warm-start* users' training ratings for training and *cold-start* users' test ratings for test. We compare with Wide&Deep network (Cheng et al., 2016), GCMC (using feature vectors) and two state-of-the-art cold-start recommendation models AGNN (Qian et al., 2019) and MeLU (Lee et al., 2019). In Table 3, we present test RMSEs for all the models, and we can see that our model IRCF-HY gives the best results, achieving 2.6% improvement over the best competitor MeLU even on the difficult zero-shot recommendation task. The result shows that our inductive model is a promising approach to handle new users with no historical behavior in real-world dynamic systems.

---

[4]In such case, we train the base MF model and inductive model over the same users in order.

## 4.2 Further Discussions

**Sparse Data and Few-shot users** A successful recommender system is supposed to handle data sparsity and few-shot users with few historical ratings. Here we construct several sparse datasets by using $50\%, 20\%, 10\%, 5\%, 1\%$ and $0.1\%$ training ratings in Movielens-1M, and then compare the test RMSEs of query users in Fig. 4(a). Also, in Fig. 4(b) we compare the test RMSEs for users with different numbers of historical ratings when under $50\%$ sparsity. As shown in Fig. 4(a), as the dataset becomes sparser,the RMSEs of all the models suffer from a drop, but the drop rate of our inductive models IGCF-NN and IGCF-GC is much smaller compared with transductive models NNMF and GCMC. In Fig. 4(b), we find that users with more historical ratings usually have better RMSE scores compared with few-shot users. By contrast, our inductive models IRCF-NN and IRCF-GC exhibits a more smooth decrease and even outperform other transductive methods NNMF and GCMC for users with very few ratings. In the extrame cases with less than five historical ratings, notably, IRCF-GC achieves $2.5\%$ improvement on RMSE compared with the best transductive method GCMC.

**Attention Weight Distribution** In Fig. 4(c) we visualize attention weights of IRCF-NN from query users to support users in Movielens-10M. As we can see, there is an interesting phenomenon that some of support users are very 'important' and most query users give high attention weights on them, which indicates that the representations of these support users are informative and a combination of them can provide powerful expressiveness for query users' preferences. In Fig. 4(d) we further plot support users' accumulated attention weights w.r.t. # historical ratings. We can see that larger attention weights concentrate on support users with more historical ratings. Such observation gives an important hint for selecting optimal support users: important support users are more likely to exist in users with more observed ratings. In Appendix F, we compare different split ways for support and query users and provide more discussions and results on this point.

**Scalability Test** We further investigate the scalability of IRCF-GC compared with two GNN-based counterparts IGMC and GCMC. We statistic the training time per epoch on Movielens-1M using a GTX 1080Ti with 11G memory. Here we truncate the dataset and use different numbers of ratings for training. The results are shown in Fig. 3 (with log-scale axis). As we can see, when dataset size becomes large, the training times per epoch of three models all exhibit linear increase. IRCF spends approximately one more time than GCMC, while IGMC is approximately ten times slower than IRCF. Nevertheless, while IRCF costs one more training time than GCMC, the latter cannot tackle new unseen users without retraining a model in test stage.

## 5 Conclusions

In this paper, we propose a new inductive collaborative filtering framework that learns hidden relational graphs among users to allow sufficient message passing in the latent space. The new model accomplishes inductively computation for user-specific representations without compromising on expressiveness and scalablity. Through extensive experiments, we show that the model can achieve state-of-the-arts performance on inductive matrix completion and outperforms transductive models for users with few training ratings. Interesting future directions include: 1) consider the selection for support users as a decision problem; 2) use meta-learning or transfer learning techniques over our inductive framework; 3) exploit our framework in cross-domain recommendation tasks.

The core idea of IRCF opens a new way for next generation of representation learning, i.e., one can consider a pretrained representation model for one set of existing entities and generalize their representations (through some simple transformations) to efficiently compute inductive representations for others, enabling the model to flexibly handle new coming entities in an open world. We believe that this novel and effective framework can inspire more researches in broad areas of AI.

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

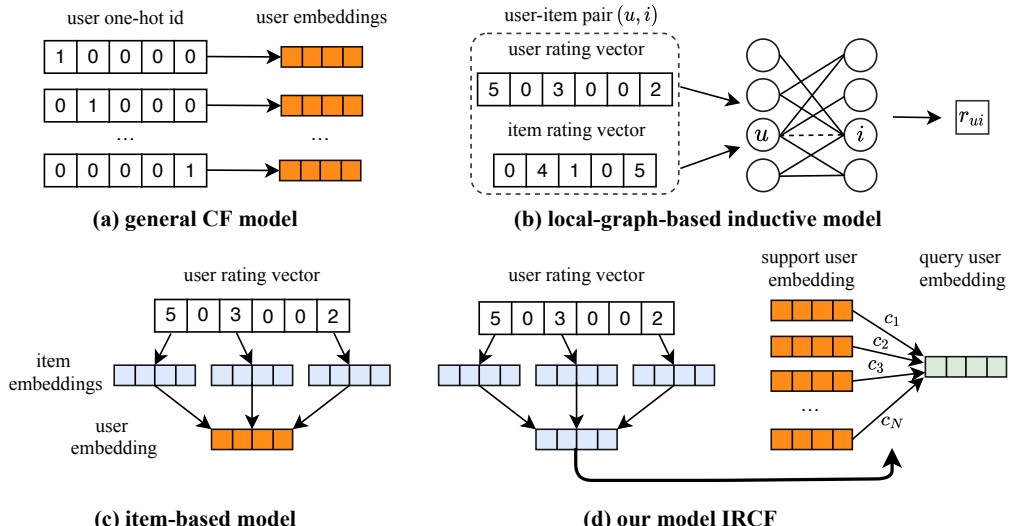

**Figure 5:** Comparison with related works on methodological level. (a) **General collaborative filtering** assumes user-specific one-hot embeddings for users and learn them collaboratively among all the users in one dataset. It disables inductive learning due to such learnable one-hot embeddings. (b) **Local-graph-based inductive model** (Zhang & Chen, 2020) extracts local subgraph structures within 1-hop neighbors of each user-item pair (i.e., rated items of the user and users who rated the item) from a bipartite graph of all the observed user-item ratings and use GNNs to encode such graph structures for rating prediction. Note that the model requires that the local subgraphs do not contain user and item indices, so it assumes no user-specific embeddings. (c) **Item-based model** leverages embeddings of user's historically rated items to compute user's embeddings via some pooling methods. The learnable parameters only lie in the item space. (d) Our model IRCF adopts item-based embedding to compute attention scores on different support users and aggregate one-hot embeddings of support users to compute user-specific embeddings for query users, which maintains ability to produce user representations with enough expressiveness and achieves inductive learning.

## A    LINKS TO RELATED WORKS

We provide a thorough discussion on the relationships and differences to related works. In Fig. 5 we present a illustration for comparison with different methods.

## B    PROOFS IN SECTION 3

### B.1    PROOF OF THEOREM 1

*Proof.* The proof is trivial by construction. Assume the optimal $\mathbf{P}_2$ for (3) as $\mathbf{P}_2^*$. Since $\mathbf{P}_1$ given by (1) is column-full-rank, for any column vector $\mathbf{p}_{u'}^*$ in $\mathbf{P}_2^*$ ($u' \in \mathcal{U}_2$), there exists $\mathbf{c}_{u'}^*$ such that $\mathbf{c}_{u'}^{*\top} \mathbf{P}_1 = \mathbf{p}_{u'}^*$. Hence, $\mathbf{C}^* = [\mathbf{c}_{u'}^*]_{u' \in \mathcal{U}_2}$ is a solution for (2) and gives $\mathcal{D}_{\mathcal{S}_2}(\hat{R}_2, R_2) < \epsilon$. $\square$

### B.2    PROOF OF THEOREM 2

*Proof.* With fixed a true rating matrix $R_2$ to be learned and a probability distribution $\mathcal{P}$ over $[M_2] \times [N]$, which is unknown to the learner, we consider the problem under the framework of standard PAC learning. We can treat the matrix $R_2$ as a function $(u', i) \to r_{u'i}$. Let $\mathcal{R}$, a set of matrices in $\mathbb{R}_{M_2 \times N}$, denotes the hypothesis class of this problem. Then the input to the learner is a sample of $R_2$ denoted as

$$\mathcal{T} = \left( (u_t', i_t, r_{u_t' i_t}) | (u_t', i_t) \in \mathcal{S}_2 \right),$$

where $\mathcal{S}_2 = \{(u_t', i_t)\} \in ([M_2] \times [N])^{T_2}$ is a set with size $T_2$ containing indices of the observed entries in $R_2$ and each $(u', i)$ in $\mathcal{S}_2$ is independently chosen according to the distribution

$\mathcal{P}$. When using $\mathcal{T}$ as training examples for the learner, it minimizes the error $\mathcal{D}_{\mathcal{S}_2}(\hat{R}_2, R_2) = \frac{1}{T_2}\sum_{(u',i)\in\mathcal{S}_2} l(r_{u'i}, \hat{r}_{u'i})$. We are interested in the generalization error of the learner, which is defined as

$$\mathcal{D}(\hat{R}_2, R_2) = \mathbb{E}_{(u',i)\in\mathcal{P}}[l(r_{u'i}, \hat{r}_{u'i})].$$

The (empirical) Rademacher complexity of $\mathcal{R}$ w.r.t. the sample $\mathcal{T}$ is defined as

$$Rad_{\mathcal{T}}(\mathcal{R}) = \frac{1}{T_2}\mathbb{E}_\sigma\left[\sup_{\hat{R}_2\in\mathcal{R}}\sum_{t=1}^{T_2}\sigma_t\hat{r}_{u'_t i_t}\right],$$

where $\sigma_t \in \{-1, 1\}$ is random variable with probability $Pr(\sigma_t = 1) = Pr(\sigma_t = -1) = \frac{1}{2}$. We assume $l$ is $L$-Lipschitz w.r.t. the first argument and $|l|$ is bounded by a constant. Then a general result for generalization bound of $\mathcal{R}$ is

**Lemma 1.** *(Generalization bound (Mohri et al., 2012)): For a sample $\mathcal{T}$ with random choice of $\mathcal{S}_2 = ([M_2]\times[N])^{T_2}$, it holds that for any $\hat{R}_2 \in \mathcal{R}$ and confidence parameter $0 < \delta < 1$,*

$$Pr(\mathcal{D}(\hat{R}_2, R_2) \leq \mathcal{D}_{\mathcal{S}_2}(\hat{R}_2, R_2) + G) \geq 1 - \delta, \tag{9}$$

*where,*

$$G = 2L\cdot Rad(\mathcal{X}) + O\left(\sqrt{\frac{\ln(1/\delta)}{T_2}}\right).$$

Based on the lemma, we need to further estimate the Rademacher complexity in our model to complete the proof. In our model, $\hat{R}_2 = \mathbf{C}^\top\mathbf{P}_1\mathbf{Q}$ and the entry $\hat{r}_{u'i}$ is given by $\hat{r}_{u'i} = \mathbf{p}_{u'}^\top\mathbf{q}_i = \mathbf{c}_{u'}^\top\mathbf{P}_1\mathbf{q}_i$ (where $\mathbf{c}_{u'}$ is the $u'$-th colunm vector of $\mathbf{C}$). Define $\mathcal{A}$ as a set of matrices,

$$\mathcal{A} = \{\mathbf{A}\in[0,1]^{M_2\times M_1}| : \|\mathbf{a}_{u'}\|_1 = \sum_{u=1}^{M_1}|a_{u'u}| = 1\}.$$

Then we have

$$T_2\cdot Rad_{\mathcal{T}}(\mathcal{R}) = \mathbb{E}_\sigma\left[\sup_{\mathbf{C}\in\mathcal{C}}\sum_{t=1}^{T_2}\sigma_t\mathbf{c}_{u'_t}^\top\mathbf{P}_1\mathbf{q}_{i_t}\right] \tag{10}$$

$$= \mathbb{E}_\sigma\left[\sup_{\mathbf{C}\in\mathcal{C}}\sum_{u'=1}^{M_2}\mathbf{c}_{u'}^\top\cdot\left(\sum_{t:u_t=u'}\sigma_t R_{1,*i_t}\right)\right] (\because R_{1,*i_t} = \mathbf{P}_1\mathbf{q}_i) \tag{11}$$

$$\leq H\cdot\mathbb{E}_\sigma\left[\sup_{\mathbf{A}\in\mathcal{A}}\sum_{u'=1}^{M_2}\mathbf{a}_{u'}^\top\cdot\left(\sum_{t:u_t=u'}\sigma_t R_{1,*i_t}\right)\right] \tag{12}$$

$$= H\cdot\mathbb{E}_\sigma\left[\sum_{u'=1}^{M_2}\max_{u\in[M_1]}\left(\sum_{t:u_t=u'}\sigma_t r_{ui_t}\right)\right]. \tag{13}$$

The last equation is due to the fact that $\mathbf{a}_{u'}$ is a probability distribution for choosing entries in $R_{1,*i_t}$, the $i_t$-th column of matrix $\hat{R}_1$. In fact, we can treat the $\max_{u\in[M_1]}$ inside the sum over all $u' \in \mathcal{U}_2$ as a mapping $\kappa$ from $u' \in [M_2]$ to $u \in [M_1]$. Let $\mathcal{K} = \{\kappa : [M_2] \to [M_1]\}$ be the set of all mappings from $[M_2]$ to $[M_1]$, and then the above formula can be written as

$$\mathbb{E}_\sigma\left[\sum_{u'=1}^{M_2}\max_{u\in[M_1]}\left(\sum_{t:u_t=u'}\sigma_t r_{ui_t}\right)\right] \tag{14}$$

$$= \mathbb{E}_\sigma\left[\sup_{\kappa\in\mathcal{K}}\sum_{u'=1}^{M_2}\sum_{t:u_t=u'}\sigma_t r_{\kappa(u'),i_t}\right] \tag{15}$$

$$= \mathbb{E}_\sigma\left[\sup_{\kappa\in\mathcal{K}}\sum_{t=1}^{T_2}\sigma_t r_{\kappa(u_t),i_t}\right] \tag{16}$$

$$\leq B\sqrt{T_2}\cdot\sqrt{2M_2\log M_1}. \tag{17}$$

The last inequality is according to the Massart Lemma. Hence, we have

$$Rad_{\mathcal{T}}(\mathcal{R}) \leq HB\sqrt{\frac{2M_2 \log M_1}{T_2}}. \tag{18}$$

Incorporating (18) into (9), we will arrive at the result in this theorem.

□

## C  SPECIFICATIONS OF IRCF

In Section 3, we present a general framework for inductive relational collaborative filtering (IRCF) without using features. In the following, we provide two different specifications for $f_\theta$ in IRCF with neural networks (IRCF-NN) and graph convolution networks (IRCF-GC) which are used in our experiments.

### C.1  NEURAL NETWORK AS BASED MODEL (IRCF-NN)

We follow the architecture in NNMF (Dziugaite & Roy, 2015) and use neural network for $f_\theta$. Here we combine a three-layer neural network and a shallow dot-product operation. Concretely,

$$f_\theta(\mathbf{p}_u, \mathbf{q}_i) = \frac{(\mathbf{p}_u^\top \mathbf{q}_i + nn([\mathbf{p}_u \| \mathbf{q}_i \| \mathbf{p}_u \odot \mathbf{q}_i]))}{2} + b_u + b_i, \tag{19}$$

where $nn$ is a three-layer neural network using $tanh$ activation, $\odot$ denotes element-wise product and $b_u, b_i$ are bias terms for user $u$ and item $i$, respectively.

### C.2  GRAPH CONVOLUTION NETWORK AS BASED MODEL (IRCF-GC)

We follow the architecture used in GCMC (van den Berg et al., 2017) and adopt graph convolution network for $f_\theta()$. Besides user-specific embedding for user $u$ and item-specific embedding for item $i$, we consider embeddings for user $u$'s rated items and users who rated on item $i$, i.e., the one-hop neighbors of user $u$ and item $i$ in user-item bipartite graph. Denote $\mathcal{N}_u = \{i | r_{ui} \neq 0\}$ as user $u$'s rated items and $\mathcal{N}_i = \{u | r_{ui} \neq 0\}$ as users who rated on item $i$. We consider graph convolution to aggregate information from neighbors,

$$\mathbf{m}_u = ReLU(\frac{1}{|\mathcal{N}_u|} \sum_{i \in \mathcal{N}_u} \mathbf{W}_q \mathbf{q}_i), \tag{20}$$

$$\mathbf{n}_i = ReLU(\frac{1}{|\mathcal{N}_i|} \sum_{u \in \mathcal{N}_i} \mathbf{W}_p \mathbf{p}_u). \tag{21}$$

Then we define the output function

$$f(\mathbf{p}_u, \mathbf{q}_i, \{\mathbf{p}_u\}_{u \in \mathcal{N}_i}, \{\mathbf{q}_i\}_{i \in \mathcal{N}_u}) = nn'([\mathbf{p}_u \odot \mathbf{q}_i \| \mathbf{p}_u \odot \mathbf{m}_u \| \mathbf{n}_i \odot \mathbf{q}_i \| \mathbf{n}_i \odot \mathbf{m}_u]) + b_u + b_i, \tag{22}$$

where $nn'$ is a three-layer neural network using $ReLU$ activation.

## D  EXTENSIONS OF IRCF

IRCF can be extended to feature-based setting and flexibly deal with zero-shot learning, as is shown in Section 4. Here, we provide details of feature-based IRCF (IRCF-HY) which indeed is a hybrid model that considers both user features and one-hot user indices. Furthermore, we discuss in the views of transfer-learning and meta-learning that can be incorporated with our framework as future study.

### D.1  HYBRID MODEL WITH FEATURES (IRCF-HY)

Assume $\mathbf{a}_u$ denotes user $u$'s raw feature vector, i.e., a concatenation of all the features (often including binary, categorical and continuous variables) where categorical features can be denoted by one-hot or multi-hot vectors. If one has $m$ user features in total, then $\mathbf{a}_u$ can be

$$\mathbf{a}_u = [\mathbf{a}_{u1} \| \mathbf{a}_{u2} \| \mathbf{a}_{u3} \| \cdots \| \mathbf{a}_{um}].$$

Then we consider user-sharing embedding function $\mathbf{y}_i()$ which can embed each feature vector into a $d$-dimensional embedding vector:

$$\mathbf{y}_u = [\mathbf{y}_1(\mathbf{a}_{u1})||\mathbf{y}_2(\mathbf{a}_{u2})||\mathbf{y}_3(\mathbf{a}_{u3})||\cdots||\mathbf{y}_m(\mathbf{a}_{um})].$$

Similarly, for item feature $\mathbf{b}_i = [\mathbf{b}_{i1}||\mathbf{b}_{i2}||\mathbf{b}_{i3}||\cdots||\mathbf{b}_{in}]$, we have its embedding representation:

$$\mathbf{z}_i = [\mathbf{z}_1(\mathbf{b}_{i1})||\mathbf{z}_2(\mathbf{b}_{i2})||\mathbf{z}_3(\mathbf{b}_{i3})||\cdots||\mathbf{z}_n(\mathbf{b}_{in})].$$

Also, we assume user-specific index embedding $\mathbf{p}_u$ and item-specific index embedding $\mathbf{q}_i$ for user $u$ and item $i$, respectively, as is in Section 3. The prediction for user $u$'s rating on item $i$ can be

$$\hat{r}_{ui} = g_\theta(\mathbf{p}_u, \mathbf{y}_u, \mathbf{q}_i, \mathbf{z}_i), \tag{23}$$

where $g_\theta$ can be a shallow neural network with parameters denoted by $\theta$. To keep notation clean, we denote $\mathbf{Y} = \{\mathbf{y}_1, \mathbf{y}_2, \cdots, \mathbf{y}_m\}$ and $\mathbf{Z} = \{\mathbf{z}_1, \mathbf{z}_2, \cdots, \mathbf{z}_n\}$. Then for support users in $\mathcal{U}_1$ with rating matrix $R_1$, we consider the optimization problem,

$$\min_{\mathbf{P}_1, \mathbf{Q}, \mathbf{Y}, \mathbf{Z}, \theta} \mathcal{D}_{\mathcal{S}_1}(\hat{R}_1, R_1), \tag{24}$$

based on which we get learned feature embedding functions $\mathbf{Y}, \mathbf{Z}$ as well as transductive embedding matrices $\mathbf{P}_1, \mathbf{Q}$ which we further use to compute inductive embeddings for query users.

For query users, feature embeddings can be obtained by the learned $\mathbf{Y}$ and $\mathbf{Z}$ in (24), i.e., $\mathbf{y}_{u'} = [\mathbf{y}_{u'1}(\mathbf{a}_{u'1})||\cdots||\mathbf{y}_{u'm}(\mathbf{a}_{u'm})]$ where $\mathbf{a}_{u'}$ is raw feature vector of user $u'$. Then we have a relation inference model $h_w$ that consists of a multi-head attention function and use user feature as input $\mathbf{d}_{u'} = \mathbf{y}_{u'}$. The inductive user-specific representation can be given by $\mathbf{p}_{u'} = h_w(\mathbf{d}_{u'})$ (i.e., (5) and (6)), similar as the CF setting in Section 3. The rating of user $u'$ on item $i$ can be predicted by $\hat{r}_{u'i} = g_\theta(\mathbf{p}_{u'}, \mathbf{y}_{u'}, \mathbf{q}_i, \mathbf{z}_i)$. Also, the optimization of the second stage is

$$\min_{w, \theta} \mathcal{D}_{\mathcal{S}_2}(\hat{R}_2, R_2). \tag{25}$$

## D.2 ZERO-SHOT LEARNING

For zero-shot recommendation where test users have no historical rating, we have no information about users if without any side information. In such case, most CF models would fail for personalized recommendation and degrade to a trivial one which outputs the same result (or the same distribution) to all the users using the popularity of items. For IRCF, the set $\mathcal{A}_{u'}$ would be empty for users with no historical rating, in which situation we can randomly select a group of support users to construct $\mathcal{A}_{u'}$ used for computing attentive scores with support users. Another method is to directly use average embeddings of all the support users as estimated embeddings for query users. In such case, the model degrades to ItemPop (using the numbers of users who rated the item for prediction).

On the other hand, if side information is available, our hybrid model IRCF-HY can leverage user features for computing inductive preference embeddings, which enables zero-shot recommendation. We apply this method to zero-shot recommendation on Movielens-1M using features in Section 4 and achieve superior RMSE.

## D.3 TRANSFER LEARNING & META-LEARNING

Another extension of IRCF is to consider transfer learning on cross-domain recommendation tasks (Singh & Gordon, 2008) or when treating recommendation for different users as different tasks like (Lee et al., 2019). Transfer learning and meta learning have shown power in learning generalizable models that can adapt to new tasks. In our framework, we can also take advantage of transfer learning (few-shot learning or zero-shot learning) or mete-learning algorithms to train our relation inference model $h_w$. For example, if using model-agnostic meta-learning algorithm for the second-stage optimization, we can first compute one-step (or multi-step) gradient update independently for each user in a batch and then average them as one global update for the model. The meta-learning can be applied over different groups of users or cross-domain datasets.

## E DETAILS IN IMPLEMENTATIONS

We provide implementation details that are not presented in Section 4 in order for reproducibility.

Table 4: Statistics of five datasets used in our experiments. Amazon-Books dataset has implicit user feedbacks while other four datasets have explicit feedbacks. Explicit feedbacks mean user's rating values range within $[1, 2, 3, 4, 5]$ and implicit feedbacks mean we only know whether a user has rated on an item or not. † A dataset has both user/item features, only user or item features, or no feature.

| Dataset | # Users/Items | # Ratings | Density | Feature† | # Supp/Query Users |
|---|---|---|---|---|---|
| Douban | 3,000/3,000 | 136,891 | 0.0152 | User | 2,131/869 |
| Movielens-100K | 943/1,682 | 100,000 | 0.0630 | User/Item | 671/272 |
| Movielens-1M | 6,040/3,706 | 1,000,209 | 0.0447 | User/Item | 5,114/926 |
| Movielens-10M | 69,878/10,677 | 10,000,054 | 0.0134 | No | |
| Amazon-Books | 52,643/91,599 | 2,931,466 | 0.0012 | No | 49,058/3,585 |

### E.1 DATASETS

The statistics of datasets used in our experiments are summarized in Table 4. We use the preprocessed versions of Douban and Movielens-100K provided by (Monti et al., 2017). For Movielens-1M and Movielens-10M [5], we use the same 9:1 training/test split as previous works. The original ML-1M dataset has a few user features and a recent work (Lee et al., 2019) collects a new dataset with more features for both users and items[6]. We use this version for our feature-based setting on ML-1M in experiments. The raw dataset for Amazon-Books [7] is a very large and sparse one and we filter out infrequent items and users with less than five ratings.

### E.2 HYPER-PARAMETER SETTINGS

We present details for hyper-parameter settings in different datasets. We use $L = 4$ attention heads for our inductive relation inference model among all the datasets. For Douban and ML-100K, each attention head randomly samples 200 support users for computing attention weights. For ML-1M and ML-10M, we set sample size as 500; for Amazon-Books, we set it as 2000. We use Adam optimizer and learning rates are searched within [0.1, 0.01, 0.001, 0.0001]. For the first-stage transductive training, we consdier L2 regularization for user and item embeddings. The regularization weights are searched within [0.001, 0.002, 0.005, 0.01, 0.02, 0.05, 0.1, 0.2]. The mini-batch sizes are searched within [64, 256, 512, 1024, 2048] to keep a proper balance between training efficiency and performance. Also, different hyper-parameters for architectures are used in three implementations.

**IRCF-NN.** For Douban and ML-100K, we use embedding dimension $d = 16$ and neural size $48 - 32 - 32 - 1$ for $f_\theta$. For ML-1M, ML-10M and Amazon-Books, we use $d = 32$ and neural size $96 - 64 - 64 - 1$ for $f_\theta$.

**IRCF-GC.** For Douban and ML-100K, we use embedding dimension $d = 32$ and neural size $128 - 32 - 32 - 1$ for $f_\theta$. For ML-1M, ML-10M and Amazon-Books, we use $d = 64$ and neural size $256 - 64 - 64 - 1$ for $f_\theta$.

**IRCF-HY.** We use embedding size $d = 32$ for each feature in ML-1M as well as user-specific and item-specific index embeddings. The neural size of $g_\theta$ is set as $320 - 64 - 64 - 1$.

### E.3 DETAILS FOR COMPARATIVE METHODS

For each comparative model in our experiments, we mainly rely on the designs and setups in their papers and, if necessary, tune its hyper-parameters to obtain optimal results in each dataset. Here we provide some implementations details for MeLU (Lee et al., 2019), AGNN (Qian et al., 2019), IMC (Zhong et al., 2019), and BOMIC (Ledent et al., 2020).

---

[5]https://grouplens.org/datasets/movielens/

[6]https://github.com/hoyeoplee/MeLU

[7]http://jmcauley.ucsd.edu/data/amazon/

- **MeLU**. We use the source codes provided by the paper[8]. The author provides two versions of the model, MeLU-1 and MeLU-5, different from using one-step or five-step local updates for each global update. We report the best results of the two in our experiments.

- **AGNN**. We build user-user (resp. item-item) graph according to cosine similarities of features and rating records of users (resp. items). In specific, we add an edge from $i$ to $j$ when $i$ is among $j$'s top 10 nearest nodes. In order for the consistency of feature dimension, we use a linear model to transform the original one-hot or multi-hot representation of each feature into a vector with fixed dimension. When training the eVAE model, we use Euclidean distance instead of cross-entropy between the original features and reconstructed features for calculating reconstruction loss.

- **IMC**. We first use a two-layer neural network to transform the multi-hot user/item features into low-dimensional vector representations. Then we calculate the dot-product of the the features of user-item pairs to make predictions.

- **BOMIC**. We use the version BOMIC+ in our experiments. To be specific, the model incorporates user features, item features, item bias, user bias as well as user/item index embeddings to predict the possible rating between users and items.

**Evaluation Metrics** We provide details for our adopted evaluation metrics. In our experiments, we follow evaluation protocols commonly used in previous works in different settings. Three metrics used in our paper are as follows.

- **AUC:** This is a measurement for consistency of recommendation list ranked by predicted scores and ground-truth clicking list with 1s before 0s. It counts the average area under the curve of true-positive v.s. false-positive curve for one user's ranking list:

$$AUC_u = \frac{\sum_{i \in \mathcal{I}_u^+} \sum_{j \in \mathcal{I}_u^-} \delta(\hat{y}_{u,i,j} > 0)}{|\mathcal{I}_u^+||\mathcal{I}_u^-|}, \tag{26}$$

where $\mathcal{I}_u^+ = \{i|r_{ui} > 0\}$ and $\mathcal{I}_u^- = \{j|r_{uj} = 0\}$ denote the sets of clicked items and not clicked items by user $u$ respectively. The indicator $\delta(\hat{r}_{ui} > \hat{r}_{uj})$ returns 1 when $\hat{r}_{ui} > \hat{r}_{uj}$ and 0 otherwise. AUC is used For implicit feedbacks. Since we only have ground-truth positive examples (clicked items) for each user, we negatively sample five items as negative examples (non-clicked items) for each user-item rating in dataset.

- **RMSE:** Root Mean Square Error measures the averaged L2 distance between predicted ratings and ground-truth ratings:

$$RMSE_u = \sqrt{\frac{\sum_{i \in \mathcal{I}_u^+} (\hat{r}_{ui} - r_{ui})^2}{|\mathcal{I}_u^+|}}. \tag{27}$$

- **MAE:** Mean Absolute Error measures the averaged L1 distance between predicted ratings and ground-truth ratings:

$$MAE_u = \frac{\sum_{i \in \mathcal{I}_u^+} |\hat{r}_{ui} - r_{ui}|}{|\mathcal{I}_u^+|}. \tag{28}$$

## F  DISCUSSIONS ON DIFFERENT CONFIGURATIONS FOR SUPPORT AND QUERY USERS

In our experiments in Section 4, we basically consider users with more than $\delta$ training ratings as $\overline{\mathcal{U}}_1$ and the remaining as $\overline{\mathcal{U}}_2$, based on which we construct support users and query users to study model's expressiveness in inductive learning and generalization on new unseen users. Here we provide a further discussions on two splitting ways and study the impact on model performance.

- **Threshold**: we select users with more than $\delta$ training ratings as $\overline{\mathcal{U}}_1$ and users with less than $\delta$ training ratings as $\overline{\mathcal{U}}_2$.

---

[8]https://github.com/hoyeoplee/MeLU

Table 5: RMSEs on all the users (ALL), query users (Query) and new users (New) of IRCF-NN in Movielens-1M using different configurations for support and query users. The results calibrate with IRCF-NN in Table 1 where we consider threshold split with $\delta = 30$ in ML-1M.

| | $\delta$ | 20 | 30 | 40 | 50 | 60 | 70 |
|---|---|---|---|---|---|---|---|
| Threshold | All (RMSE) | 0.8440 | 0.8437 | 0.8439 | 0.8440 | 0.8444 | 0.8451 |
| | Query (RMSE) | 0.9785 | 0.9525 | 0.9213 | 0.9166 | 0.9202 | 0.9160 |
| | New (RMSE) | 0.9945 | 0.9912 | 0.9902 | 0.9883 | 0.9911 | 0.9929 |
| | $\gamma$ | 0.97 | 0.85 | 0.75 | 0.68 | 0.62 | 0.57 |
| Random | All (RMSE) | 0.8446 | 0.8536 | 0.8587 | 0.8637 | 0.8669 | 0.8689 |
| | Query (RMSE) | 0.8863 | 0.8848 | 0.8760 | 0.8805 | 0.8824 | 0.8855 |
| | New (RMSE) | 0.9901 | 0.9923 | 0.9956 | 0.1001 | 1.0198 | 1.0262 |

- **Random**: we set a ratio $\gamma \in (0, 1)$ and randomly sample $\gamma \times 100\%$ of users in the dataset as $\overline{\mathcal{U}}_1$. The remaining users are grouped as $\overline{\mathcal{U}}_2$.

We consider $\delta = [20, 30, 40, 50, 60, 70]$ and $\gamma = [0.97, 0.85, 0.75, 0.68, 0.62, 0.57]$ (which exactly gives the same ratio of $|\overline{\mathcal{U}}_1|$ and $|\overline{\mathcal{U}}_1|$ as corresponding $\delta$ in threshold split[9]) in Movielens-1M dataset. For each spliting, we also consider two situations for support users and query users: 1) $\mathcal{U}_1 = \overline{\mathcal{U}}_1$ and $\mathcal{U}_2 = \overline{\mathcal{U}}_2$; $\mathcal{U}_1 = \mathcal{U}_2 = \overline{\mathcal{U}}_1$. The results of IRCF-NN are presented in Table 5 where we report test RMSEs on all the users, query users and new users.

As we can see from Table 5, in threshold split, as $\delta$ increases (we have fewer support users and more query users and they both have more training interactions on average), test RMSEs for query users exhibit a decrease. The reason is two-folds: 1) since support usres have more training ratings, the transductive model can learn better representations; 2) since query users have more training ratings, the inductive model would have better generalization ability. On the other hand, with different spliting thresholds, test RMSEs for new users remain in a fixed level. The results demonstrate that our model performance on new unseen users is not sensitive to different split thresholds. However, in random split, when $\gamma$ decreases (also we have fewer support users and more query users but their average training ratings stay unchanged), RMSEs for new users suffer from an obvious decrease. One possible reason is that when we use smaller ratio of support users in random split, important support users in the dataset are more likely to be ignored. (As is shown in Fig. 4(c) in Section 3, there exist some important support users that give high attention weights for query users.) If such support users are missing, the performance would be affected due to insufficient expressive power of inductive model.

Comparing threshold split and random split, we can find that when using the same ratio of support users and query users (i.e., the same column in Table 5), RMSEs on new users with threshold split are always better than those with random split. Such observation shows that support users with more training ratings would be more effective for inductive learning and again accord with the results in Fig. 4(d) which demonstrates important support users who give large attention weights for query users tend to exist in users with many training ratings.

---

[9]For example, using $\gamma = 0.97$ in random split will result in the same sizes of $\overline{\mathcal{U}}_1$ and $\overline{\mathcal{U}}_2$ as using $\delta = 20$ in threshold split.

