# OpenReview forum: "Inductive Collaborative Filtering via Relation Graph Learning"
_ICLR.cc/2021/Conference — Reject_

### Official Review · AnonReviewer2 · 2020-10-25
**This is a borderline paper and slightly above the threshold**

**Rating:** 6
**Confidence:** 4

**Review:**

This paper proposed an inductive collaborative filtering method, called IRCF. The goal is to possess expressiveness (against feature-driven methods) as well as generalization (against one-hot encoding based methods). In IRCF, there are a matrix factorization model for support users and a relation model for query users. The former is trained with transductive learning to obtain support users embeddings and item embeddings. The relation model then generates query user embeddings as weighted sum of support user embeddings by examining relational graph between support and query users.

Pros:

1. The paper is well-written and easy to follow.
2. The experimental results are satisfying.
3. The Theorem 1 and Theorem 2 reflect the tradeoff between capacity and generalization, which can guide the way of selecting support users.
4. The idea of using a set of pretrained embeddings as bases may be generalized to other inductive tasks.

Cons/Questions:

1. A detailed related work section is expected. There have been many works studying inductive recommendation problems w/ or w/o user features. As far as I known, lots of methods like FISM generate user embeddings by aggregating embeddings of historical items, which naturally support inductive learning. In this paper, the proposed method IRCF views query user embeddings as weighed sum of support user embeddings, but the weights are still based on aggregating embeddings of historical items (i.e., d_u' in Eq. 4). Moreover, the support users are analogous to a set of bases, and each user can be represented by a combination of the bases. Thus, it is hard to assess the novelty without a related work section.
2. It is unclear on how to handle user bias terms b_u in Eq. 19 and Eq. 22 for query users or new users.
3. It seems that you assume C is a conical combination coefficients in Eq.4. Why not to use the unnormalized scores in Eq. 4, which matches the Theorem 1 better?

---

> ### Author Response · Authors · 2020-11-16
> **We add a thorough discussion on differences to related works in Section 2**
>
> Thanks for your comments and review. On the methodological side, the novelty of our model lies in jointly learning a global graph over users and user representations based on attention weights in the graph. Such an idea brings up several advantages over previous works. 1) Compared with inductive matrix completion with features or with item-based embedding (like FISM [1]), our model maintains superior capacity using both learnable parameters in user and item space. 2) Compared with inductive matrix completion with local-graph structures [2], our model possesses better expressiveness and scalability (justified in Section 3.2 and our experiments). 3) Compared with transductive one-hot embedding-based CF models, our model achieves inductive learning and does not sacrifice any capacity (justified by Thm 1 and our experiments). In short, IRCF, as a new inductive matrix completion model, unifies the advantages of other methods and overcomes their limitations in one general CF framework.
>
> Q1: Related works section
>
> We add several paragraphs in Section 2 to discuss relationships and differences to other papers (including general CF, feature-driven models, inductive matrix completion, item-based models). Also, we add a figure in Appendix A as comparison with other works.
>
> Q2: How to handle user bias term
>
> For query users (whose training ratings used for learning inductive models), we learn the b_u together with model optimization. For new users (unseen by our model during training), we use averaged values of other users’ bias as an estimation.
>
> Q3: Unnormalized attention scores
>
> We tried using an unnormalized version in our experiments before, and the performance is not good compared to the normalized scores. Also, we empirically found that normalized scores can help to stabilize the training and avoid mode collapse on a small group of support users
>
> References:
>
> [1] Kabbur et al., FISM: factored item similarity models for top-n recommender systems. In KDD 2013.
>
> [2] Zhang et al., Inductive Matrix Completion Based on Graph Neural Networks. In ICLR. 2020.

---

### Official Review · AnonReviewer4 · 2020-10-28
**An interesting paper with limited novelty and solid technical solution**

**Rating:** 6
**Confidence:** 5

**Review:**

### Quick summary
This work explores a popular problem, i.e., collaborative filtering, in an inductive setting, which is very important for real-world recommender systems. To address the challenges in the inductive settings, i.e., learning accurate representations for users who do not occur in the training data, the authors propose to construct a relational graph between users in the training data and new users based on a standard matrix factorization model and then use an attentive message passing framework to inductively compute user-specific representations. Besides, the authors prove the expressive and generalization capabilities of the proposed framework. Extensive experiments are conducted to demonstrate the effectiveness of the proposed framework both in transductive and inductive settings, as well as the scalability.

### Clarity
The presentation of the paper is good.

### Originality
Generally speaking, the inductive collaborative filtering is of limited novelty, while the technical solution is novel and solid, especially the part in constructing a relational graph between support users and query users.

### Pros
1. The technical solution is interesting and solid, with a clear presentation in the paper.
2. The proofs of Theorem 1&2 are interesting, which theoretically shows the expressiveness and generalization abilities of the proposed model.
3. The experiments are extensive, most of which are convincing.
4. The presentation of the paper is good.

### Cons
1. The major concern is that the novelty of inductive collaborative filtering with GNN is limited since Zhang & Chen 2020 [1] proposed the IGMC framework, which has done a comprehensive study on the inductive CF problem. Though the authors point out the difference between the proposed IRCF and Zhang's work, they do not give adequate materials to support their arguments. For example, the mentioned disadvantage of IGMC is that *the subgraphs in IGMC are ignorant of user and items indices*, however, from the perspective of the author, this issue is not that important, and may not occur very frequently, and can be trivially addressed by incorporating the user and item indices into IGMC. It will be more convincing if the authors can give more supporting materials in the paper.
2. For Theorem 1, the authors hold one argument that matrix factorization gives maximized capacity for learning personalized user preferences from historical rating patterns, however, it does not make sense to the reviewer. Can you provide any references or explain a bit more? Besides, what are the implications of Theorem 1 in helping us understanding the proposed IRCF ?
3. For Theorem 2, the authors only discuss the influences of the size of $\mathcal{U}_1$. How about other variables, e.g., $B, H, M_2$, etc.
4. In the performance comparisons, the authors use RMSE in Table 1 and 2, while MAE in Table 3. This seems weird to the reviewers. Why do not you adopt the same metric, say either RMSE or MAE, since the experiments are actually the same type.

Generally speaking, the paper is of high quality. The idea is clear, and the technical solution is interesting and solid with a theoretical guarantee. Most experimental results are convincing. Besides, the writing of the paper is clear and easy to understand.

------
### Post rebuttal
Great thanks to the authors for the detailed replies. After reading them, I decided to keep my rating.

---

> ### Author Response · Authors · 2020-11-16
> **The novelty of our model lies in jointly learning graph structures and node representations to enable inductively computing user embeddings**
>
> We thank for your review and comments. Here is a response to the questions.
>
> Q1: Comparison with IGMC
>
> We need to point out that the rationale of IGMC is totally different from our model, though both consider inductive matrix completion without features and use GNNs as tools. IGMC extracts a local subgraph of 1-hop neighbors for a user-item pair and uses GNNs to encode such subgraph to get predicted rating value. Note that such subgraphs only contain structure information without user and item indices, and this is why IGMC can achieve inductive learning (if incorporating user index, the model has to learn one-hot user embedding/representation and become limited in transductive learning). By contrast, our IRCF first learns matrix factorization for users and extends user embeddings for new users via message passing over estimated hidden relational graphs. In short, there are two main differences between two models. 1) IGMC cannot produce user embeddings/representations while IRCF maintains such ability in inductive learning. The user embeddings are important for user profile representation and many downstream tasks, like target advertisement, user-controllable recommendation, influence maximization, etc. Also, IRCF as an embedding-based model possesses superior expressiveness than local-graph-based model (IGMC), as shown in Section 3.2 and our experiments.  2) IGMC focuses on subgraph structures in an observed bipartite graph, while IRCF considers learning a hidden global graph over users. The joint learning of graph learning and graph representation can address noisy and incomplete information (resulted from user exposure bias) in observed user-item ratings.
>
> Q2: more discussions for Thm 1
>
> Since matrix factorization assumes a learnable embedding vector for each user, such one-hot embeddings provide maximized capacity for learning user’s preferences compared to shared embeddings in other models (e.g., feature-driven model with common feature space or item-based models with common item space). Thm 1 indicates that IRCF can minimize the loss to the same level as standard matrix factorization under one mild condition, which shows that our proposed inductive model does not sacrifice any model capacity.
>
> Q3: more discussions for Thm 2
>
> B and H can be treated as constants and have small effects. B denotes the bound for user’s rating on items, determined by specific datasets, e.g. B=5 in movielens. H is determined by sparsity of the hidden relational graph. In most cases, we found the hidden graph is very sparse (as shown in Fig. 4(c)) so H is not a large value in practice. M2 is the number of query users. The generalization error bound becomes tighter with fewer query users, in which case the model tends to focus more on specific users in the context of collaborative learning among query users.
>
> Q4: different metrics
>
> We agree that using the same metric would be more convincing. In Table 1 and 2, we follow the common benchmarks used in NNMF, GCMC, IGMC, F-EAE, using RMSE as metrics in Movielens and Douban datasets. In Table 3 for experiments on new users, we follow the strong competitor MeLU in cold-start recommendation, directly use its provided datasets (a feature-augmented version for ML-1M) and follow the evaluation protocol in its paper using MAE as metric. We update the results of RMSE for Table 3 experiment in the paper. The results are consistent with MAE. IRCF achieves 0.9367 and exceeds MeLU with 0.9625 over  a large margin.

---

### Official Review · AnonReviewer1 · 2020-11-03
**vote to reject**

**Rating:** 4
**Confidence:** 5

**Review:**

Summary:
The work proposes a relational learning scheme that extends standard collaborative filtering approach and aims at improving recommendations quality for new users. The proposed method extracts relations from historical rating data within a preselected subset of support users and utilizes this knowledge to better represent newly introduced (query) users.

Reasons for score:
The paper misses comparison with a large set of competitive techniques based on incremental learning approach and provides no justification for this omission. Evaluation methodology seem to have test data leak, which may be the major source of performance gains instead of the model itself.

Pros:
The authors present an interesting view on relational learning problem within the collaborative filtering setting. Generating recommendations online in an instant fashion for both known and new users is indeed a very relevant problem of a great practical importance. The proposed relational learning-based modification of standard collaborative filtering schemes is described clearly and incorporates novel ideas. The authors also prove two theorems, which further support feasibility of their approach and provide some hints on the expected behavior.

Cons:
Both in the abstract and in the text, the authors state that standard CF techniques are unable to deal with new users without the need to retrain the entire model. This statement is wrong. For example, in the case of warm start scenario with matrix factorization techniques one can easily update the model specifically for a newly introduced user by performing a few update steps using only ratings of this user. In the case of SGD this would be just a few "half-steps" of gradient descent with fixed matrix of item embeddings. This can be generalized to neural-networks based approaches. Similarly, for the ALS algorithm (e.g., [1]) it would require "half-step" of solving a linear system w.r.t. new user embedding (see eq. 4 in [1]), which can be performed efficiently and in fact is one of the standard approaches in many production systems.
Furthermore, this can be even reduced to an analytical solution in the case of PureSVD approach [2]: it only requires learning item embeddings, and the user embeddings are simply represented as a weighted sum of the embeddings of items they interacted with (see eq. 6 in [2]). It's similar to the way the d_u variable is defined in this paper in eq. 4. Finally, a natural generalization of the latter representation would be an autoencoder, e.g. MultVAE [3] or RecVAE [4], which naturally resolves the warm-start scenario without the need for any modifications as there's no need for a separate user representation.
Therefore, in order to make comparison complete I would suggest to include some incremental learning techniques as well as autoencoder solutions and clearly demonstrate how the proposed method compares to them in terms of recommendation quality, flexibility, and computational efficiency.

The second major point here is the evaluation methodology. In eq. 4, matrices W_q, W_k represent trainable parameters. So how are they trained? Equation 7 explicitly states that training is done on historical data from query users (matrix R_2) and no other data splits are present.  If that's the case, than there's a test data leak: historical rating data of query users is used to train parameters of the model and then the same users are used to evaluate the performance of the model. It doesn't correspond to the warm start scenario. The weights W_q, W_k must be fixed first (after they were trained) and then used to generate representations of users that were never shown to the model before with eq. 6. Hence, there should be at least two disjoint subsets of query users: one for validation and another one for actual testing of the model. Unfortunately, I couldn't find any hints on such a splitting neither in the main text, nor in the Appendix, which makes me believe there's no such splitting. Avoiding test data leaks is absolutely crucial for a fair comparison.

Minor comments:
The problem of generating recommendations is not the same as the problem of rating prediction/matrix completion. It's important to keep this distinction in mind. The standard task for recommender systems is generating an ordered list of relevant items. The quality of this cannot be measured with RMSE. In fact, there's a strong evidence that models that perform well in terms of RMSE metric may not be good at all in terms of more appropriate metrics like precision, recall, nDCG, MAP, MRR, etc. Please, consider adding more appropriate metrics into the work.
Also note that AUC is not the best choice for that matter as it makes no distinction between proper ranking of irrelevant items and proper ranking of relevant items. Considering that the majority of items in recommendations are typically irrelevant, high AUC scores may not reliably represent actual performance of algorithms in their ability to generate lists of relevant items.

References:
[1] Hu, Yifan, Yehuda Koren, and Chris Volinsky. "Collaborative filtering for implicit feedback datasets." In 2008 Eighth IEEE International Conference on Data Mining, pp. 263-272. Ieee, 2008.
[2] Cremonesi, Paolo, Yehuda Koren, and Roberto Turrin. "Performance of recommender algorithms on top-n recommendation tasks." In Proceedings of the fourth ACM conference on Recommender systems, pp. 39-46. ACM, 2010.
[3] Liang, Dawen, Rahul G. Krishnan, Matthew D. Hoffman, and Tony Jebara. "Variational autoencoders for collaborative filtering." In Proceedings of the 2018 World Wide Web Conference, pp. 689-698. 2018.
[4] Shenbin, Ilya, Anton Alekseev, Elena Tutubalina, Valentin Malykh, and Sergey I. Nikolenko. "RecVAE: A New Variational Autoencoder for Top-N Recommendations with Implicit Feedback." In Proceedings of the 13th International Conference on Web Search and Data Mining, pp. 528-536. 2020.

---

> ### Author Response · Authors · 2020-11-16
> **Clarifications for misunderstandings and more discussions**
>
> Thanks for your comments. We need to point out that there are two types of situations widely considered in recommender system area: 1) explicit feedback dataset with user-item rating information (e.g.  1,2,3,4,5);  2) implicit feedback dataset with user-item click (or view/like) information (e.g. 0/1). In the first case the recommendation task is usually formalized as a matrix completion problem [4-7], and RMSE as performance metric is a common practice in this setting. In the second case the top-N recommendation setting is widely used for evaluation. Our paper mainly focus on the first case, so we use RMSE as evaluation metric in Douban and Movielens datasets (with explicit feedbacks), following our competitors [4-7]. Also, we agree that the top-N recommendation is a very important and realistic problem, and we leave it as a focus of future works. In our experiments in Amazon dataset (with implicit feedbacks), we can also use top-N metrics (see Q4 in the following).
>
> Q1: Incremental learning for CF models
>
> We change the statement “has to retrain the whole/entire model” to “has to retrain the model” which can be more precise. In fact, the incremental learning is a decent choice for new users in online systems. However, it also requires model learning for each new user (gradient descent for neural models or computing matrix inversion for linear systems with ALS algorithm), limiting the efficiency in inference. Also, such incremental learning is designed independently for each user, which would be more prone for over-fitting than collaborative learning among users in CF models. We experiment incremental learning on ML-100K/ML-1M, and get RMSE 1.014/0.9608 on query users. Our IRCF gives RMSE 0.981/0.944 (in Table 1), which are significantly better.
>
> Q2: Comparison with item-based models
>
> We add discussions of relationships and differences to related works (including item-based models [1][2], VAE models [3]) in Section 2. Admittedly, such item-based and VAE models can inductively compute user embeddings without the need to retrain a model for new users. However, they have very limited capacity for learning user preferences, since they only consider learnable parameters in item space. By contrast, IRCF considers both learnable parameters in user and item space, with enough capacity as equivalent as general matrix factorization (which gives state-of-the-art performance on matrix completion). In fact, these methods are not used as baselines in the experiments of our main competitors NNMF, GCMC, IGMC, F-EAE. As further demonstration, we implemented [1], [2], [3] and get test RMSEs 2.920/2.090 for [1], 2.276/1.911 for [2], 2.981/2.861 for [3] on new users in ML-100K/ML-1M. Our IRCF gives test RMSEs 0.999/0.956, which exceed them by a large margin.
>
> Q3: Test data leak issue
>
> This is definitely a misunderstanding for our evaluation. We strictly follow training/validation/test split for evaluation. In fact, as indicated in our experiment setup (1st paragraph in Section 4 and 1st paragraph in 4.1), we split training/test ratings for all the users in one dataset. Then we use the number of training ratings for each user to split users into two sets $ \overline {\mathcal{U}}_1$ and $ \overline {\mathcal{U}}_2$. We consider two situations: 1) use $ \overline {\mathcal{U}}_1$ as support users and $ \overline {\mathcal{U}}_2$ as query users; 2) use $ \overline {\mathcal{U}}_1$ as both support and query users. In both cases, we use training ratings of support (resp. query) users to train our transductive (resp. inductive) model. In the first case, we report RMSEs for test ratings for all (resp. query) users which corresponds to All (resp. Query) in Table 1. In the second case, we report RMSE for test ratings of users in $ \overline {\mathcal{U}}_2$ (new users unseen by the model) which corresponds to New in Table 1. Our experiments provide a fair comparison with other baselines in various situations.
>
> Q4: Evaluation metrics
>
> In Amazon dataset, if we consider Recall@3/Recall@10 , we get 0.251/0.670 for IRCF-GC, 0.243/0.650 for IRCF-NN, 0.226/0.590 for NNMF, 0.223/0.588 for GCMC. The results are consistent with AUC results in Table 2 and our IRCF achieves the best performance.
>
> References:
>
> [1] Cremonesi et al. "Performance of recommender algorithms on top-n recommendation tasks." In Proceedings of the fourth ACM conference on Recommender systems, pp. 39-46. ACM, 2010
>
> [2] Kabbur et al. "FISM: factored item similarity models for top-n
> recommender systems." In KDD, 2013.
>
> [3] Liang et al., "Variational autoencoders for collaborative filtering." In WWW 2018.
>
> [4] Dziugaite et al., “Neural Network Matrix Factorization.” CoRR, abs/1511.06443. 2015.
>
> [5] Berg et al., “Graph Convolutional Matrix Completion.” CoRR, abs/1706.02263. 2017.
>
> [6] Zhang et al., “Inductive Matrix Completion Based on Graph Neural Networks.” In ICLR 2020.
>
> [7] Hartford et al., “Deep Models of Interactions Across Sets.” In ICML 2018.

---

### Official Review · AnonReviewer5 · 2020-11-06
**Interesting work with some caveats**

**Rating:** 6
**Confidence:** 4

**Review:**

##########################################################################

Summary:

This work proposed an inductive recommendation framework on user-item relation graphs. Such a framework relies on the user-item relations without the requirement of side-information and perceives certain flexibility in terms of the parametrization for user/item representations. The authors also provided theoretical analysis to highlight some mathematical insights out of this framework. The proposed method is evaluated on three real-world datasets and compared with several baselines.

Overall I find the work was well-reasoned and executed in a relatively good shape, thus recommending acceptance.


##########################################################################

Strength:
- Relevant topic to the ICRL community and could have potential impact in real-world applications
- The proposed method is well reasoned and technically sound
- Experiments are executed in a decent shape

##########################################################################

Weakness:
- Motivations behind its technical contributions can be further sharpened; comparisons to previous related studies on the inductive graph learning domain can be further improved
- Some gaps between the current experiment setup and real-world recommendation senarios

##########################################################################

Detailed Comments:

I'll address the above potential weakness in details here.

I personally find a bit difficult to digest the motivations of this work and how it differentiated from previous inductive graph learning work until diving into its detailed parametrizations. Fig. 1 and its descriptions are helpful in terms of illustrating the inductive setting, but not quite informative in terms of concrete contributions of this work conceptually.
My takeaway from the proposed framework is, the attentive pooling method falls into the aggregator family of inductive graph learning, despite that the aggregation and sampling scheme are performed on user side globally instead of on the user-item local neighborhoods. In this regard, it may also be helpful to highlight the (mathematical) difference between this work and existing inductive graph learning (e.g. pinSage) after eq.5/6.

Although the experimentations are executed in a good shape, there are still some gaps between the current setup and real-world recommendation requirements.
- The proposed method is largely evaluated on the rating prediction setting, AUC is reported on the amazon dataset but no Top-K ranking metrics are performed during the experiments. It is acceptable given these metrics are consistent with the optimization objective, however, the notable gap between pointwise prediction setting and the real-world online top-K ranking setting needs to be called out.
- Another concern about the current evaluation protocol is, it enforces the temporal dynamics on the user side and assumes item representations remains the same - again it is consistent with the proposed method (i.e., Q remains the same) thus expected to favor it. The question is, whether these assumptions are consistent with real-world senarios. As far as I know, both movieLens and Amazon datasets have associated timestamps, what the real temporal dynamics here and what would be the warm/cold item/user distribution look like if splitting data chronologically?


Minor Concerns:
	- Annotations in  Figure 4 can be further enlarged for visibility

---

> ### Author Response · Authors · 2020-11-16
> **We add discussion for motivations and differences to inductive graph representation learning**
>
> Thanks for your review and comments. We address the proposed issues in order.
>
> Q1: Motivations of our model
>
> The motivation lies in three-folds
> 1) Conceptually, in many recommender systems, user behaviors and preferences share a lot of proximity. For one user, his/her behaviors would be impacted by a group of other users. Based on these observations, we can model user’s behaviors/preferences via combination of other users’.
> 2) Mathematically, in matrix factorization framework, user’s embedding can be expressed as a weighted combination of base vectors which span the d-dimensional latent space. If support users’ embeddings are full-column-rank (as in thm 1), we can leverage them as base vectors to express arbitrary embeddings for query users.
> 3) From the perspective of graph representation learning, the observed graph often contains noisy links or missing important links. In our case, if we directly use the bipartite graph of user-item ratings to define a user-user graph, the graph would be pretty sparse especially for new users and miss potential links (ratings) due to user exposure bias. Hence, we turn to jointly learning graph structures (as attention scores) and node representations through graph attentive convolution, which brings up better accuracy and flexibility.
>
> Q2: Difference to inductive graph representation works
>
> IRCF jointly estimate neighbored nodes for a target node (given by attention scores) and learn node representations based on that, while inductive graph representation models often assume a given observed graph and directly learn node representations. Furthermore, IRCF can deal with nodes with new users with no historical edge, while the latter would fail for new nodes with no observed edge (if without node attribute features). We update this discussion after eqn 5/6 in our paper. Also, we summarize differences to other related works in Section 2 and add a figure in Appendix A for conceptual illustration.
>
> Q3: Experimental evaluation
>
> We adopt RMSE as metrics in most of our experiments since most of baseline methods in our paper (NNMF, GCMC, IGMC, F-EAE, etc.) consider RMSE as the only metric in douban, ML-100k, ML_1M, ML-10M datasets. We use the same benchmarks to calibrate with them. Our paper focus on CF framework for general matrix completion task. It would be interesting to extend our method to ranking-based top-N recommendation as future work. Also, in Amazon dataset with implicit feedbacks, we can also consider top-N metrics for evaluation. For example, if we consider Recall@3/Recall@10 , we get 0.251/0.670 for IRCF-GC, 0.243/0.650 for IRCF-NN, 0.226/0.590 for NNMF, 0.223/0.588 for GCMC. The results are consistent with AUC results in Table 2 and our IRCF achieves the best performance.
>
> Q4: Temporal dynamics for users
>
> It is a misunderstanding for our evaluation. We do not assume temporal dynamics on user side. For different rated items of a user, say $(u, i_1, r_1, t_1)$ and $(u, i_2, r_2, t_2)$ where $t_1$, $t_2$ denote different time when the rating behavior happens. We drop the timestamps and assume the same user historical set $\mathcal I_u$ for both rating instance. In other words, we do not consider time information in our model and experiments, and assume all the ratings with no temporal order. Such settings are widely adopted in our comparative methods (e.g., GCMC, IGMC) and other papers for general recommendation.

---

### Decision · Program_Chairs · 2021-01-07
**Final Decision**

**Decision:**

Reject

**Comment:**

The paper is somewhat borderline, though reviews mostly lean positive. Unfortunately after calibrating compared to other submissions, the work remains somewhat below the bar compared to higher-scoring papers.

The reviewers praise the topic, the method, and the experiments (although some of this praise is a little mixed or lukewarm). The most negative review raises several specific concerns about the evaluation methodology, as well as some concerns about data leaks etc. While serious, the authors rebuttal to these claims seems reasonably convincing. While the remaining issues appear not to be dealbreakers, there are nevertheless some lingering concerns which ultimately put the paper slightly below the bar.

The AC notes that their initial inclination was to accept this paper, though it was suggested that the score be lowered after calibration compared to other submissions, mainly due to doubt regarding these lingering issues.